# The First Snake Venom KTS/Disintegrins-Integrin Interactions Using Bioinformatics Approaches

**DOI:** 10.3390/molecules28010325

**Published:** 2022-12-31

**Authors:** Oussema Khamessi, Hazem Ben Mabrouk, Selim Kamoun, Chaima Hkimi, Kais Ghedira, Riadh Kharrat

**Affiliations:** 1Laboratoire des Venins et Biomolécules Thérapeutiques, Pasteur Institute of Tunis, University of Tunis El Manar, 13 Place Pasteur BP 74, Tunis 1002, Tunisia; 2Institut de Biotechnologie de Sidi Thabet, Université de la Manouba, Ariana BP-66, Manouba 2010, Tunisia; 3Laboratory of Bioinformatics, Biomathematics and Biostatistics (LR20IPT09), Pasteur Institute of Tunis, University of Tunis El Manar, Tunis 1002, Tunisia

**Keywords:** KTS-disintegrins, anti-tumor, integrin, bioinformatics, protein-protein docking, normal modes analysis

## Abstract

Snake venom contains a number of active molecules that have been shown to possess high anti-tumor activities; disintegrins are an excellent example among these. Their ability to interact and bind with integrins suggests that they could be very valuable molecules for the development of new cancer therapeutic approaches. However, in the absence of a clear Lysine-Threonine-Serine (KTS) Disintegrins Integrin interaction model, the exact compound features behind it are still unknown. In this study, we investigated the structural characteristics of three KTS-disintegrins and the interaction mechanisms with the α1β1 integrin receptor using in silico bioinformatics approaches. Normal mode analysis showed that the flexibility of the KTSR motif and the C-terminal region play a key role and influence the KTS-Disintegrin-integrin interaction. Protein-protein docking also suggested that the interaction involving the KTSR motif is highly dependent on the residue following K21, S23 and R24. These findings contribute to a better understanding of the KTS-Disintegrin-Integrin structural differences and their interactions with α1β1 receptors, which could improve the selection process of the best active molecules for antitumor therapies.

## 1. Introduction

In the past few years, natural substances have witnessed an exponentially growing therapeutic interest in many fields, particularly venoms, which are natural sources of biomolecules with potential therapeutic effect against cancer [1]. Bioinformatics has become essential for identifying new drug candidates. Understanding a protein’s structure can reveal a lot about how it works, and this is especially true given the exponential increase in the number of novel sequences produced each year. A first strategy is based on the homology principle. Indeed, homologous proteins with significant sequence similarity have a very similar three-dimensional structure and may share the same biological function [2]. In that regard, many bioinformatics tools have been developed in order to help and assist in protein structures and interactions understanding. Homology modelling is regarded as the most accurate computational structure prediction approach for determining a protein’s 3D structure from its amino acid sequence, making it a crucial approach for the development of new therapeutic candidates. In addition to modelling individual proteins, it can also help predict protein interactions [3]. Bioinformatics, one of the most recent interdisciplinary fields involving the study and processing of biological information using computer science and mathematical methods, has become an essential complement to traditional biology’s classic in situ, in vivo, and in vitro approaches [4]. In this light, the identification, isolation, and characterization of active compounds originating from natural sources (by targeting many receptors overexpressed or selectively expressed in cancer cells) has immense therapeutic potential [5,6]. Several studies have shown that the disintegrins derived from snake venom are also endowed with significant anti-cancer effects [7]. The KTS-disintegrins, which are a family of small, non-enzymatic proteins, represent a good example [8]. The first KTS disintegrin (short monomeric molecules of 41 AA) was discovered in the venom of *Vipera lebetina obtusa* and was called Obtustatin [9]. Three disintegrins have the functional motif KTS: Obtustatin, Viperistatin and Lebestatin. These molecules interact specifically with integrins alpha1, beta1 (α1β1) receptors, expressed mainly on the surface of vascular endothelial cells as antagonists of adhesion to their native ligand (type IV collagen), thereby inhibiting the angiogenesis of cancer cells and tumor growth [10]. According to Arnaout et al., integrin-ligand binding requires the presence of divalent cations, with a typical preference for manganese, magnesium, and calcium (Mn+2, Mg+2, and Ca+2), although the relative preference for optimal affinity varies between the different heterodimers [11,12]. The KTS motifs bind to the collagen receptor of α1β1 integrin in a strong and specific manner at the level of the Mg+2 metal ion, inhibiting integrin binding to its natural ligand [13,14]. Despite the fact that Obtustatin, Viperistatin, and Lebestatin are three high homologous disintegrins that block the Domain-I of α1 integrin (DIα1), these molecules seem to block cell adhesion and migration on collagen IV in distinct ways [15]. Indeed, their inhibition activity is quite different, with IC50s of roughly 2, 0.2, and 0.08 nM on collagen IV for Obtustatin, Lebestatin and Viperistatin, respectively [16]. Currently the explanation behind the difference in activity among these molecules is still unclear. From this general point of view, we have approached this problem with an in silico study. Herein, we performed a computational analysis based on bioinformatics approaches in order to highlight the structural differences between these three molecules and their interactions to α1β1 integrin. In the present work, we attempt to better understand the KTS-Disintegrin-Integrin structural differences and their interactions with α1β1 receptors, which could improve the selection process of the best active molecules for antitumor therapies.

## 2. Results

### 2.1. Bioinformatics Approaches

The analysis reported here was performed using a computational approach based on the combination of computational analysis, molecular modelling, normal modes analysis, and molecular docking. Figure 1 highlights the different steps used in the present study to investigate the structure differences between KTS/disintegrins molecules and the molecular docking with the DIα1 of the α1β1 integrin.

### 2.2. Disintegrins Phylogenetic Tree Analysis

The evolution pathway of disintegrin structure diversification involved the reduction of the polypeptide chain and the selective loss of pairs of cysteine residues that form disulfide bonds [17,18,19]. Phylogenetic analysis suggests that short disintegrins represent the more recent diverging lineages of disintegrins. The main goal of this study is to show the evolutionary relationships between the different disintegrin subfamilies. The disintegrins phylogenetic tree displayed in Figure 2, including the group of short disintegrins corresponding to Arginine-Glycine-Aspartic (RGD) group, the RGD like group (include Lysine-Glycine-Aspartic (KGD), Methionine-Leucine-Aspartic (MLD), Methionine-Glycine-Aspartic (MGD), Valine-Glycine-Aspartic (VGD), and Tryptophan-Glycine-Aspartic (WGD)), and the Arginine/Lysine-Threonine-Serine (R/KTS) group.

This short disintegrins family containing 41–51 residues and four disulfide bonds, has revealed three clades corresponding to (RGD), (KGD), and R/KTS motifs with, respectively, 40, 5, and 5 disintegrins (50 disintegrins in total).

### 2.3. Molecular Modelling

Sequence alignment revealed the existence of multiple highly conserved residues and few variable ones, as well as the absence of gaps (Figure 3D). Viperistatin and Lebestatin have both displayed a high level of identity (92.7% and 95.1%, respectively), when compared to Obtustatin. Despite the few differences between their sequences, there are significant disparities in activity in terms of inhibiting the adhesion of cancer cells, in relation to their adhesion sites.

In order to investigate and decipher this latter, we performed a molecular modelling study of the Viperistatin and Lebestatin. The homology between these peptides is reinforced by the presence of eight cysteine cross-linked by four disulfide bridges for each one and the presence of several conserved segments, which probably play a structural role, in particular the KTS motif on the N-terminal side, which 3D structure has shown a similar fold for the three peptides (Figure 3A–C). The similarity at the level of the primary structure, which reaches 95%, is confirmed at the level of the tertiary structure.

### 2.4. Normal Modes Analysis

Normal modes analysis has shown that the C-terminal end is flexible for these three disintegrins and therefore does not have a major influence on the overall structure of the molecule (Figure 4A). However, Viperistatin and Lebestatin exhibit greater structural flexibility than Obtustatin (Figure 4B). To characterize the flexibility of KTS-disintegrins, we calculated the B-factor values per Cα atom for each disintegrin. The correlation values estimated by Elnemo are of 0.777, 0.698, and 0.672, for the low-energy of Obtustatin, Lebestatin, and Viperistatin, respectively. The best RMSD is computed with respect to the second conformer. <R2> gives a visualization of the mean square displacement (NMSD) of all C-alpha atoms associated with a given mode. In general, the NMSD/R2 profile analysis shows that some residues tend to be more mobile than others, especially around residues 23, 24, and 41 (with 0.04 cut-off values). The NMSD profile suggests that the N-terminals of the KTS-disintegrin are stable (Figure 4B). 

### 2.5. Calculation of Solvent Accessible Surface Areas

The accessible surface calculated for each residue, using GETAREA, shows a minimum exposure for Obtustatin residues and a maximum exposure for Lebestatin and Viperistatin residues (Table 1). For Obtustatin, Lebestatin, and Viperistatin, the accessible surface area calculated using GETAREA shows a minimum of exposure for Lys21 (39.7, 17.1, and 35%), Thr22 (22, 27.2, and 34.3%), and Val^38^/Ser^38^/Leu^38^ (0.0%), and a maximum (100%) for Ser23 (100, 84.3, and 84%), Arg/Leu24 (98.4, 96.2, and 91.4%) and Gly41.

This study confirms that Viperistatin and Lebestatin possess a greater solvent accessibility of the integrin binding loop than Obtustatin. Indeed, Ser23, Arg24, and Gln41 present high values over 90%.

### 2.6. Molecular Docking (Protein-Protein Docking)

KTS/disintegrins molecules were analyzed to identify their interactions with the receptor (DIα1 of the integrin α1β1). The examination of the protein-protein docking suggests that the members of KTS-disintegrin family (Obtustatin, Lebestatin, and Viperistatin) present a critical residue (KTSR loop and other regions) able to interact with the DIα1 of the integrin α1β1. We evaluated the interaction between Lebestatin and DIα1. In our case, this interaction covers the largest part of the binding site of the collagen. Six amino acids of the Lebestatin are involved in the interaction with the interface of the DIα1. Indeed, R8, Q9, P14, and S23 are able to interact with Y17, R148, Q80, and H118. Two positively charged residues (R24 and H27) are able to interact with N120 and Q80 (with 1.9 A° and 2.6 A°, respectively). Our molecular docking study shows that neither the T22 residue nor the residues at position 38 (V38/S38/L38) and 40 (Q40/P40) are involved in the binding of KTS-disintegrins with their target (Figure 5A).

The mode of interaction predicted for the Obtustatin shows that the S23 and G41 are involved in the interaction with the residues surrounding the MIDAS site of the DIα1 (N14 and R79 with 2.7 A° and 2 A°, respectively). Figure 5B shows that T17 and the segments W20 and K21 interact with N150 and R148, respectively.

Several amino acid residues in Viperistatin make direct contact with the C-helix segment (144GSYNR148) and the residues surrounding the collagen binding site of the DIα1 (Figure 5C). The segment 18–23 (T18, C19, W20, K21, S23) binds to the MIDAS site (N14, R49, D114, E116, and S117); the residues G16, R24, and H27 interact with the C-helix segment of DIα1 (R148, N150, E159) with 2.8 A°, 2 A°, and 2.3 A°, respectively.

## 3. Discussion

Integrins play important roles in cancer physiopathology and tumor cell development, regulating a myriad of cellular functions [20]. Snake venom consists of a variety of proteins that were reported to be able to interact with these integrins, conferring them a potential role in terms of new therapeutic alternatives to cancer treatment [21,22]. Among these molecules, we investigated Obtustatin, Viperistatin, and Lebestatin, three homologous disintegrins belonging to the KTS-family that block the integrin’s α1β1 in various ways. Obtustatin, Lebestatin, and Viperistatin notably blocked the adhesion of integrin α1β1 with IC50 values of 2, 0.2 and 0.08 nM, respectively [8,11,13,23,24]; meanwhile, Viperistatin has the highest inhibiting cell adhesion activity on collagen IV [25]. This variation could be the result of the residue substitutions, thus impacting the disintegrins activities. 

This original study highlights the discovery of the first KTS/disintegrins-integrin interactions. Despite their KTS-family sequence homology (Figure 2), the disintegrins phylogenetic tree shows that three clades correspond to RGD, KGD, and R/KTS motifs and two groups include MLD and WGD motifs (41 to 51 AA). The significant activity of these molecules binding affinity reveals that these disintegrins have distinct structural characteristics (Figure 3). Moreover, Figure 3D shows that Lebestatin varies from Obtustatin at positions R^24^/L^24^ and S^38^/L^38^, as well as Viperistatin at locations R^24^/L^24^, S^38^/L^38^, and P^40^/G^40^. Due to the lack of an established interaction model of α1β1 snake-venom disintegrin, their interaction mechanisms are yet to be investigated. In the present study, we performed an in silico analysis to highlight the structural differences between these three molecules and their interactions to α1β1. 

Bioinformatics is a powerful tool that, correctly used, can provide valuable structural information to understand the conformations of individual molecules as well as the interactions between interacting molecules. However, like any predictive technique, bioinformatics also has weaknesses that particularly affect the verification of the predictions. 

In the present work we performed a computational protein-protein docking analysis based in order to understand the structural differences between three KTS-disintegrin molecules and their cognate receptor, the collagen IV binding α1β1 integrin. 

To this end, conformational ensembles of the KTS-disintegrins (ligand) and the DIα1 as a receptor were used for the cross-docking. In the literature, the interaction between Obtustatin, Viperistatin, and Lebestatin with the domain αI of integrin α1β1 domains is based on experimental data suggesting that ^21^KTS^23^ are directly involved in the binding activity of these disintegrins [26]. However, in a previous investigation, it was hypothesized that the snake venom disintegrins interaction with α1β1 may not be limited to the KTS triplet, and that other key residues may be involved in the complex’s formation. Viperistatin and Lebestatin, which exhibited higher inhibitory effects on cell adhesion and migration, also showed the presence of an Arginine positively charged side chain instead of the Leucine one in position 24. That could be linked to the formation of a better bond with the receptor’s residues. Indeed, from a fundamental structure standpoint, the exposed residues to the solvent that are directly involved in the interaction with the DIα1 have proven to be distinct across the three disintegrins (Table 1). For Obtustatin, Lebestatin, and Viperistatin, the accessible surface area calculated using GETAREA shows a minimum of exposure for Lys21 (39.7, 17.1, and 35%), Thr22 (22, 27.2, and 34.3%) and Val^38^/Ser^38^/Leu^38^ (0.0%) and a maximum (100%) for Ser23 (100, 84.3, and 84%), Arg/Leu24 (98.4, 96.2, and 91.4%) and Gly41. In light of these findings, we investigated whether flexibility at the C-Terminal level may be another crucial element in disintegrin-integrin interactions (Figure 4). Residues at the C-terminal end of the sequence cause a high instability. Figure 4B shows that the G41 residue at the C-terminal end of the sequence causes a high instability for Obtustatin. 

The analysis of the Elastic Network Model for the KTS-disintegrins profile explains this result well (Figure 4B). In order to avoid redundancy, we did not use molecular dynamics because Daidone et al. (2012) have already carried this out. In that regard, this work served here as a confirmation for that previous study, and the normal mode analysis makes it possible to decipher the flexibility that influences the interaction (described previously). The molecular dynamics simulations of the two polypeptides in aqueous solution showed that Lebestatin possesses a higher flexibility of the C-terminal tail, and a greater solvent accessibility of the integrin binding loop, compared to the Obtustatin. It was hypothesized that these properties may contribute to the higher binding-affinity of Lebestatin to its biological partner [10]. 

To answer this hypothesis, we resorted to the analysis of normal modes as well as molecular docking in order to bring more light concerning the difference in activity (experimental one) despite the strong homology between these disintegrins. In this study, normal modes analysis combined with molecular docking were used to investigate the structural characteristics of three disintegrins (Obtustatin, Lebestatin, and Viperistatin) and the interaction mechanisms with the α1β1 integrin receptor. Viperistatin has numerous residues (8 AA) on the interaction surface that engage directly with its natural ligand (type 4 collagen) via the α domain (Figure 5B). Figure 4C shows that six residues (6 AA) are involved in the ligand-receptor interaction for Lebestatin. Its fixation, however, differs from that of Viperistatin, since it is shifted from the Mg2+ fixation site (MIDAS) at the α1β1 integrin level. After filtering clustering steps and molecular visualization, Lebestatin and Viperistatin present a critical Arginine residue (Arg24) that projects into the DIα1. On the other hand, compared to Viperistatin and Lebestatin, focusing on the entire surface of the binding site Obtustatin has less residues interacting with it (5 AA), making its fixation on the DIα1 domain a completely distinct process (Figure 5A).

Protein-protein docking also suggested that the interaction of KTS-disintegrins-integrins involving the KTSR motif is highly dependent on the residue following Lys21, Ser23, and Arg24. The interaction mode between Obtustatin and DIα1 lacks the R24 amino acid, which establishes a tight interaction with the receptor. This result supports previous findings that the Obtustatin substitution for ^21^KTSX^24^ slightly decreased the inhibitory activity of the mutant, whereas the L24R mutation increased the potency of the mutant by 6-fold compared to wild-type Obtustatin [9,19,24]. The substitution of Leu24 by an Arginine residue leads to an increase in coordination of the Mg+2 ion in the MIDAS center. It should also be noted that Obtustatin lacks the Arg24 involved in the interaction and its predictions are perfectly correlated with the experimental results [9].

Our protein-protein docking study shows the critical implication of the KTSR segment in the interaction with the integrin. Furthermore, the results pointed out the fact that the Thr22 residue does not seem to intervene in the interaction, given the fact that it is buried in the ^21^KTS^24^ loop. On the other hand, the Lys21, Ser23, and Arg24 residues have a much greater probability of interaction with the target receptor. Previous studies have hypothesized that residues at position 38 and 40 are involved in the interaction [10]. We have shown that the mutations of residues 38 and 40 at the level of Lebestatin and Viperistatin have no influence on the interaction. This is also the case for residue 38, which does not play a major role in the interaction process, as it is not completely exposed to the solvent. These three disintegrins molecular docking results are in accordance with the solvent accessible surface areas analysis. It is important to note here that regarding the residue Gly41, for both Viperistatin and Lebestatin, this one does not actively participate in the interaction with the α1β1 receptor, and this, despite having a 100% exposure to the solvent. However, unlike the two other disintegrins, Obtustatin Gly41 residue plays an important role in ensuring the interaction’s stability regarding the Obtustatin-DIα1 complex, through the formation of a 2 A° hydrogen bond with the Arginine 79 (Figure 5A). 

This result is compliant with the fact that Obtustatin displays a high flexibility level for Gly41 compared to Viperistatin and Lebestatin. Indeed, previous studies have also indicated that the low flexibility in the 38–41 segment is essential to stabilize the structure of the 20–24 residue. Normal modes analysis (NMA) analysis has shown that the C-Terminal region is flexible for the three disintegrins, and therefore might not have a major impact on the overall structure of the molecule. Beyond Ser23 residue, the profile shows us that this region is flexible and NMSD/R2 shows that some residues tend to be more mobile than others. Especially with regards to the residues in position Ser23, Arg24, and Gly41, this investigation has confirmed the direct involvement of its residues in the interaction with the target receptor. 

For example, residue 41 of Obtustatin exhibits high flexibility and the latter is involved in the interaction. The Ser23 residue of Viperistatin and Lebestatin show high flexibility compared to Obtustatin. This residue is not involved in the interaction of the Obtustatin-DIl1 complex. Obtustatin lacks the Arg24 involved in the interaction and its predictions are perfectly correlated with the Figure 4B. Thr22 residue does not seem to intervene in the interaction and disintegrins molecular docking results are in accordance with the NMA. We would like to note that it is possible to find flexible regions that contain stable residues, or the reverse is due to protein conformations. This is one of the folding problems, for example, the synthetic RGD motif (cilengitide) is composed of two residues R and D that are 100% exposed to the solvent. On the other hand, the residue G is not exposed to the solvent and even carries a percentage less than 20%; this is the case for several proteins especially the ones that carry an active site.

However, both Viperistatin and Lebestatin had a more flexible C-terminal end and increased solvent accessibility more than Obtustatin, suggesting that other structural characteristics could play a key role in the differential activity of these disintegrins. This could indicate that these properties, while not being the only major ones, could prove to be of high importance for the interaction with the small hydrophobic pocket located on the surface of the integrin. This could explain the higher binding-affinity of Viperistatin and Lebestatin to their biological ligands, despite the high homology level they share with the Obtustatin.

## 4. Materials and Methods

### 4.1. Sequences Extraction and Computational Analysis

The whole analysis was carried out using a server with Operating system Linux—Ubuntu 20.04 and Memory/RAM 16 GB RAM. The uniprot sequence search has already been conducted in this present work to identify all the members belonging to this family, which were then analyzed (the phylogenetic analysis). All sequences of the KTS/disintegrins family (Obtustatin, Lebestatin, and Viperistatin) were extracted from the Uniprot database under the accession numbers P83469, Q3BK14 and P0C6E2, respectively. “This KTS/disintegrins family is composed of 41–51 residues and four disulfide bridges”.

The homologous sequences were retrieved using the protein basic local alignment search tool (BLAST: https://blast.ncbi.nlm.nih.gov/Blast.cgi, 14 January 2022). Briefly, the protein data bank proteins (RCSB PDB) was selected as a subject database [27] and the PSI-Blast (Position-Specific Iterated BLAST) algorithm was selected to be used for sequence comparison between disintegrins and the PDB database [28]. In order to exhibit common patterns that play a functional role, the MSA of disintegrins was determined by MAFFT (Multiple Alignment using Fast Fourier Transform) v7.490 [29].

### 4.2. Phylogenetic Tree Determination

The neighbor-joining technique [30] was used to infer evolutionary history. The evolutionary distances were computed using the JTT matrix-based method and are in the units of the number of amino acid substitutions per site [31]. A gamma distribution (shape parameter = 1) was used to represent rate variance among sites. The evolutionary analyses were conducted in MEGA X [32]. 

### 4.3. Lebestatin and Viperistatin Molecular Modelling

The selection of 3D models is based on the sequence homologous level between the target protein and its template. The amino acid sequences of Lebestatin and Viperistatin were compared with other sequences retrieved from nrNCBI database using FASTA and BLAST. The RMN structure of Obtustatin (PDB code 1MPZ contain 22 conformer), in which the best 3D structure (conformer) was used as a template to build the 3D structure of Lebestatin and Viperistatin using homology modelling with modeler version 10.2 [33]. Geometric evaluations were performed by Ramachandran plot using PROCHECK [34] and stereochemical quality of the modelled 3D structures of Lebestatin and Viperistatin were performed using different servers, such as the Prosa II Z-score and Verify3D [35,36].

### 4.4. Docking Ensemble Preparation

Computational protein-protein docking analysis based in order to understand the structural differences between three KTS-disintegrin molecules and their cognate receptor, the collagen IV-binding α1β1 integrin. To this end, conformational ensembles of the KTS-disintegrins (ligand) and the DIα1 as a receptor were used for the cross-docking. Docking is performed over the entire surface of the integrin. The interface of the receptor used for this molecular docking study covers the binding site of the collagen.

The first step of the docking ensemble preparation involves two PDB entries of the DIα1 domain of α1β1 integrin (receptor) used to perform our computational study. The conformational ensemble for the receptor, was obtained using two structures of the DIα1 domain corresponding to PDB entries 1PT6 (X-ray crystallography structure) and 2M32 (NMR structure) representing the bound and the unbound state for a collagen mimetic peptide, respectively [26,37]. The selected structures of the receptors correspond to the lowest energies belonging to predefined local minimums [38]. All conformational ensembles of the ligand were constructed through homology modelling.

### 4.5. Normal Modes Analysis (NMA)

Among the main computational techniques to study protein dynamics are molecular dynamics (MD) and normal modes analysis (NMA). Applications of NMA (normal mode analysis) cover wide areas of structural bioinformatics. Normal mode analysis does not take into account solvent interactions explicitly and entropic contributions to binding interactions. In our case, we used normal mode analysis just to characterize the flexibility of KTS-disintegrins. We calculated the B-factor values per Cα atom for each disintegrin. “The main goal of the Normal Mode analysis in this study is to characterize the flexibility of KTS-disintegrins (ligand)” and to confirm the previous study [10] suggesting that the KTS-disintegrins had a more flexible C-terminal end and an increased solvent accessibility compared to the Obtustatin.

These computational methods can be used to explore conformational space [39]. The models generated were then classified according to their structural similarity and the ones retained are generally the centers of the clusters [40]. Once the set of models has been generated, the program selects those with the best scores to refine them. The refining enabled it to transition from a phase of “low resolution” structures to one of atomic details, in which the atoms of the side chains are added in rigid blocks. Each generated model that has the correct score will be used as a starting point. The template that has the highest homology score with the proteins to be modelled is then selected. 

For KTS-disintegrins analysis, we used ElNémo “a normal mode for protein movement analysis” in order to improve the exploration of the ligand’s (Obtustatin, Viperistatin, and Lebestatin) flexibility [41]. ElNémo determines the normal modes that contribute the most to the corresponding protein movement when two conformations of the same (or a homologous) protein are available. “The main goal of the Normal Mode analysis in this study is to characterize the flexibility of KTS-disintegrins and to confirm the previous study “structural and dynamical properties of KTS-disintegrins” suggesting that the KTS-disintegrins had a more flexible C-terminal end and an increased solvent accessibility compared to the Obtustatin” [10].

ElNémo is a Web-interface (http://www.sciences.univ-nantes.fr/elnemo, 5 June 2022) to the Elastic Network Model (ENM), a fast and simple way for computing the low frequency normal modes of a macromolecule [42], that also has the advantage of being able to treat both small/medium size proteins as well as the very large ones. ElNémo is able to generate the 100 lowest-frequency modes and then produces a comprehensive set of descriptive parameters and visualizations, such as the degree of collectivity of movement, residue mean square displacements, distance fluctuation maps and the correlation between observed and normal-mode-derived atomic displacement parameters “B-factors”. If only a homologue of the reference protein (<100% sequence identity) is available in a different conformation, ElNémo computes the RMSD between the normal mode perturbed models and the homologous structure in order to identify the normal mode perturbations that best describe the associated protein movement.

### 4.6. Calculation of Solvent Accessible Surface Areas

This step consists of predicting the suspected active residues (hotspots) of the ligand’s that are involved in the KTS-disintegrins-Integrin interaction. Calculation of Solvent Accessible Surface Areas (SASA) allows us to identify the residue exposed to the solvent. We used the GETAREA method of calculating the SASA implemented in program fantom [43].

### 4.7. Molecular Docking

Molecular docking was performed using the ClusPro web server (https://cluspro.org/, 21 June 2022). This approach makes it possible to use experimental data (biological and/or biophysical) deduced from mutagenesis or nuclear magnetic resonance (NMR) experiment [44]. ClusPro is able to select the centers of highly populated clusters of the low energy structures rather than simply the lowest energy conformations as predictions of the complex. The docking surface is restricted to the extracellular surface of the DIα1 domain of α1β1 integrin. The docking was conducted by imposing the three disintegrins KTS-family (ligand) and the receptor’s binding site from the receptor’s structure as interacting segments. ClusPro places the DIα1 domain of Integrin at the origin of the coordinate system on a fixed grid, the KTS-disintegrin is placed on a movable grid, and the interaction energy is written in the form of a correlation function. Conformational ensembles of the KTS-disintegrins (ligand) and the DIα1 as a receptor were used for the cross-docking. The receptor binding site contains the C-helix segment (144GSYNR148) and the residues surrounding the collagen-binding site on the DIα1 domain. The collected results were organized based on the energy classification. ClusPro is a web based server for the direct docking of two interacting Proteins. The 10 best models generated by ClusPro were retained. All docking model images were generated using Pymol “The PyMOL Molecular Graphics System” [45].

## 5. Conclusions

Obtustatin, Lebestatin, and Viperistatin represent the shortest known snake venom monomeric disintegrins. In the present study, we used several in silico approaches, ranging from molecular modeling and docking to normal mode analysis, in order to understand the structural differences between three KTS-disintegrin molecules and their cognate receptor, the collagen IV-binding α1β1 integrin. Our study contributes to a better understanding of the structural distinctions between the different KTS-disintegrins and their interactions with α1β1 receptors, which could highly improve the first snake venom KTS/disintegrins-integrin interactions using bioinformatics approaches.

## Figures and Tables

**Figure 1 molecules-28-00325-f001:**
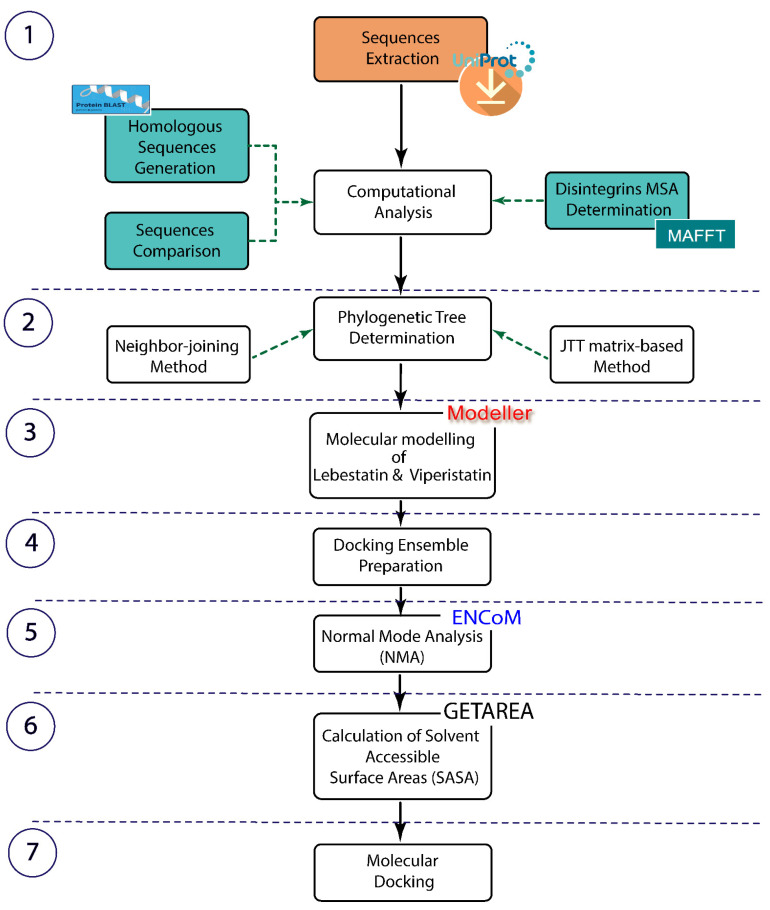
An overview highlighting step by step the present study to investigate KTS/disintegrins-integrin interactions using bioinformatics approaches.

**Figure 2 molecules-28-00325-f002:**
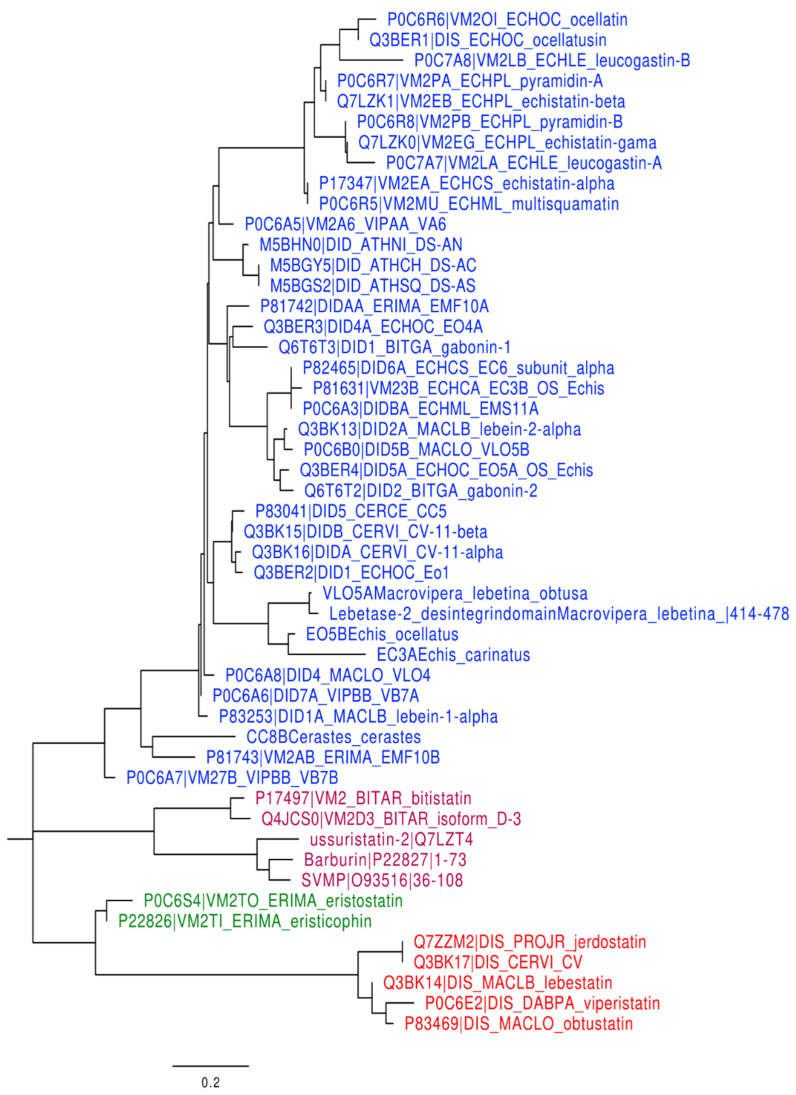
Phylogenetic tree of the disintegrins family. Phylogenetic analyses by JTT matrix-based method of all disintegrins family based on the RGD, KGD and R/KTS motifs. This analysis involved 50 amino acid sequences. Topology was supported by 1000 bootstrap replicates. The red color highlights the KTS-disintegrin family. Short disintegrins divided into groups, the color blue highlights the RGD group (24 disintegrins) and the RGD like group (14 disintegrins) includes seven MLD, five KGD, one MGD, four VGD and two WGD, Eristostatin group (green), KGD group (pink) and/RKTS (red).

**Figure 3 molecules-28-00325-f003:**
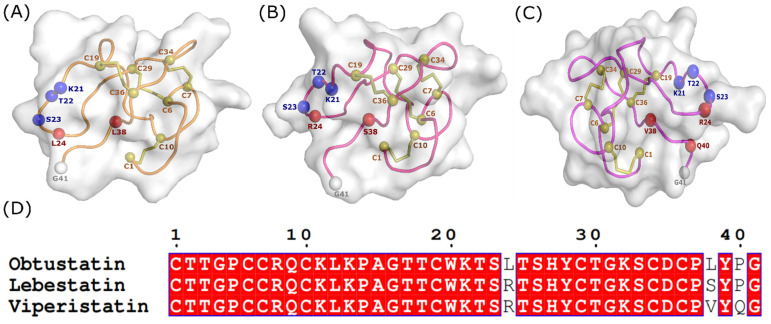
Homology modeling of KTS-disintegrin. KTS motif represented in blue, the disulfide bridge (Cn) stained in olive, the C-terminus (G41) colored in white and all mutations colored in red. (**A**) Obtustatin NMR structure (code PDB 1MPZ); (**B**) 3D structure of Lebestatin; (**C**) Viperistatin generated by homology modeling; and (**D**) multiple alignment sequences of the KTS-disintegrins family.

**Figure 4 molecules-28-00325-f004:**
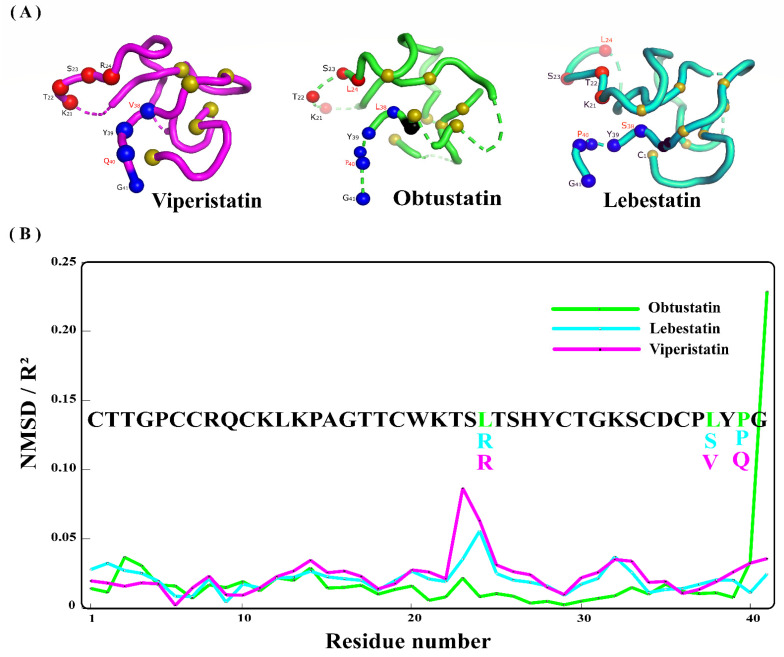
The Elastic Network Model for the KTS-disintegrins. (**A**) The individual normal modes analysis (animations, RMSD) for Viperistatin, Lebestatin, and Obtustatin. Visual inspection of the 3D structure shows the importance of the flexibility in C-terminals end. (**B**) Illustration of how the conformation changes between the KTS-disintegrins using the best low-frequency modes.

**Figure 5 molecules-28-00325-f005:**
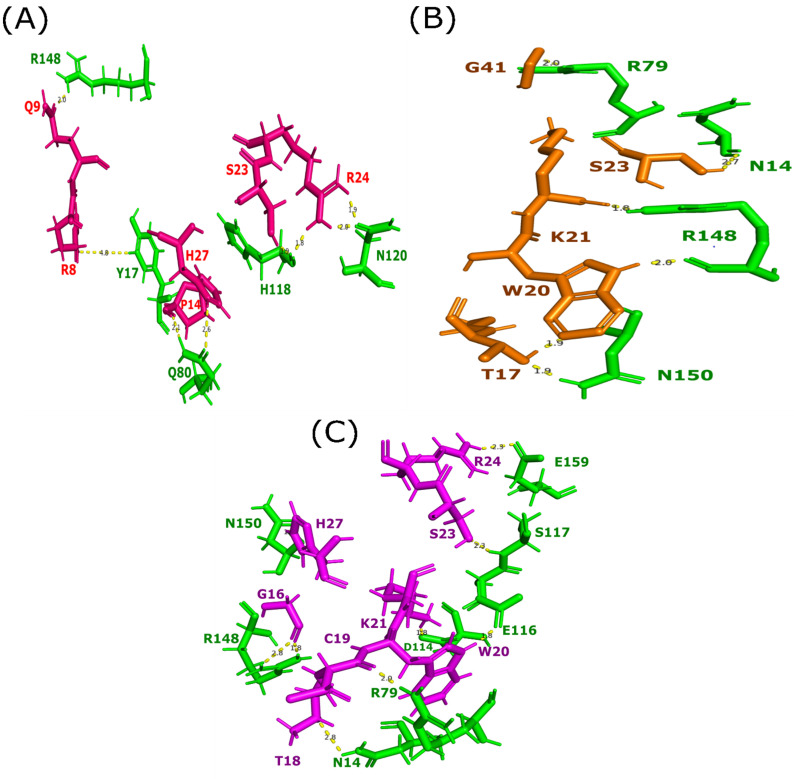
Protein-protein docking of KTS-family with domain I of the integrin α1β1 Three dimensional structures of complexes KTS-disintegrin showing the binding sites disintegrin with the DIα1 (green) and the main amino acid residues implicated in the interaction (Pink, Olive, or Magenta). (**A**) Interactions formed by 6-amino acids of Lebestatin (Pink) within the domain I of α1β1. (**B**) Docking complex of Obtustatin-DIα1 showing that 5-amino acids of the Obtustatin (olive) interact with the residues (green) surrounding the MIDAS site of the I-domain. (**C**). Interactions formed by 8-amino acids of Viperistatin (magenta) within domain I of α1β1.

**Table 1 molecules-28-00325-t001:** Solvent accessible surface areas.

	K21	T22	S23	L/R24	V/S/L38	G41
Viperistatin	39.7%	22%	100%	98.4%	0%	100%
Lebestatin	17.1%	27.2%	84.3%	96.2%	0%	100%
Obtustatin	35%	34.3%	84%	91.4%	0%	100%

## Data Availability

Data are available on request from the authors.

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
