# Peer review of "The First Snake Venom KTS/Disintegrins-Integrin Interactions Using Bioinformatics Approaches"

_molecules, 2022, doi:10.3390/molecules28010325_

Round 1

Reviewer 1 Report

The presented manuscript provides an insight into the molecular interaction between KTS-disintegrin and integrin using computational tools such as multiple sequence analysis, normal mode analysis, and molecular docking. There is a good flow of information, and the methodology is described clearly. However, I believe that the technical approach used in this study is weak and needs to incorporate molecular dynamics simulations, free energy calculation, and RMSD analysis of bound KTS-disintegrin with integrin to explain the stability of the bound complex and interaction mechanisms between KTS-disintegrin and integrin along with the suggested corrections before publishing this study.

Major concerns:

1.       A normal mode analysis does not take into account solvent interactions explicitly and entropic contributions to binding interactions. Therefore, I recommend incorporating molecular dynamics simulations into this study and comparing the binding affinity to experimental data.

2.       It would be helpful to understand the stability of the binding complex by analyzing the RMSD of the KTS-disintegrin bound to integrin.

3.       A 2D plot of the interaction between KTS-disintegrin and integrin would provide a better insight into the interaction mechanism.

4.       Rewrite the conclusion of the study as it appears superficial.

Minor Concerns:

  1. Was the DIα1 domain of Integrin fixed in docking? Have docking experiments been conducted at a specific binding site or have blind docking experiments been performed? 
  2. There is no explanation of the full form of the terms KTS, RGD, KGD, MLD, WGD, and RMN.
  3. Was the author interested in investigating compound features or mechanisms behind the interaction (line 14)?
  4. The title of Figure 3 indicates that the KTS motif is stained blue. I disagree with the use of the term "stained" here. A stain is a chemical dye used to color a specific chemical constituent for microscopy. As an alternative, I would call it a "KTS motif represented in blue."
  5. A reference to the previous study is missing from the text (line 372).

Author Response

Dear Reviewer,

Below we provide the point-by-point responses. All modifications in the manuscript have been highlighted in red. We hope that this carefully revised manuscript meets your high standards.

Best regards

Response to Reviewer 1

Major concerns:

Point 1: A normal mode analysis does not take into account solvent interactions explicitly and entropic contributions to binding interactions. Therefore, I recommend incorporating molecular dynamics simulations into this study and comparing the binding affinity to experimental data.

Response1 : We completely agree with the reviewer. Normal mode analysis does not take into account solvent interactions explicitly and entropic contributions to binding interactions. But in our case, we used normal mode analysis just to characterize the flexibility of KTS-disintegrins. We calculated the B-factor values per Cα atom for each disintegrin.  “The main goal of the Normal Mode analysis in this study is to characterize the flexibility of KTS-disintegrins (ligand)” and to confirm the previous study “structural and dynamical properties of KTS-disintegrins” suggesting that the  KTS-disintegrins had a more flexible C-terminal end and an increased solvent accessibility compared to the Obtustatin”.

We thank the reviewer for this comment. We have modified the paragraph to improve clarity.

Lines 386 “Applications of NMA (normal mode analysis) cover wide areas of structural bioinformatics. Normal mode analysis does not take into account solvent interactions explicitly and entropic contributions to binding interactions. In our case, we used normal mode analysis just to characterize the flexibility of KTS-disintegrins. We calculated the B-factor values per Cα atom for each disintegrin.  “The main goal of the Normal Mode analysis in this study is to characterize the flexibility of KTS-disintegrins (ligand)” and to confirm the previous study [10] suggesting that the KTS-disintegrins had a more flexible C-terminal end and an increased solvent accessibility compared to the Obtustatin.”

Normal mode analysis does not take into account solvent interactions explicitly and entropic contributions to binding interactions. But in our case, we used normal mode analysis just to characterize the flexibility of KTS-disintegrins. We calculated the B-factor values per Cα atom for each disintegrin.  “The main goal of the Normal Mode analysis in this study is to characterize the flexibility of KTS-disintegrins (ligand)” and to confirm the previous study [10] suggesting that the  KTS-disintegrins had a more flexible C-terminal end and an increased solvent accessibility compared to the Obtustatin”.

Point 2: It would be helpful to understand the stability of the binding complex by analyzing the RMSD of the KTS-disintegrin bound to integrin.

Response 2 : We thank the reviewer for this comment. Molecular docking using ClusPro is useful to understand the stability of the binding complex by analyzing the RMSD of integrin-bound KTS disintegrin. ClusPro performs three computational steps as follows: (1) rigid body docking by sampling billions of conformations, (2) root-mean-square deviation (RMSD) based clustering of the 1000 lowest energy structures generated to find the largest clusters that will represent the most likely models of the complex, and (3) refinement of selected structures using energy minimization for each scoring scheme, using pairwise root mean square deviation (RMSD) as the distance measure. ClusPro outputs the centers of the 10 largest clusters.  While the minimization generally removes potential steric clashes, it does not substantially change the conformation of the complexes, and thus the RMSD of our ClusPro submissions from the native complexes is fully determined by the rigid body docking and clustering steps.

Point 3: A 2D plot of the interaction between KTS-disintegrin and integrin would provide a better insight into the interaction mechanism.

Response 3: We agree with the reviewer. We have modified the figure.5 and the paragraph to improve clarity.

Lines 173 “We evaluated the interaction between Lebestatin and DIα1. In our case, this interaction covers the largest part of the binding site of the collagen. Six amino acids of the Lebestatin are involved in the interaction with the interface of the DIα1. Indeed, R8, Q9, P14 and S23 are able to interact with Y17, R148, Q80 and H118. Two positively charged residues (R24 and H27) are able to interact with N120 and Q80 (with 1.9A° and 2.6A°, respectively). Our molecular docking study shows that neither the T22 residue nor the residues at position 38 (V38 /S38 /L38) and 40 (Q40 /P40) are involved in the binding of KTS-disintegrins with their target (Figure 5A). The mode of interaction predicted for the Obtustatin shows that the S23 and G41 are involved in the interaction with the residues surrounding the MIDAS site of the DIα1 (N14 and R79 with 2.7A° and 2A°, respectively). Figure 5B shows that T17 and the segments W20 and K21 interact with N150 and R148, respectively. Several amino acid residues in Viperistatin make direct contact with the C-helix segment (144GSYNR148) and the residues surrounding the collagen binding site of the DIα1 (Figure 5C). The segment 18-23 (T18, C19, W20, K21, S23) binds to the MIDAS site (N14, R49, D114, E116 and S117); the residues G16, R24 and H27 interact with the C-helix segment of DIα1 (R148, N150, E159) with 2.8 A°, 2A° and 2.3A°, respectively.”

Point 4: Rewrite the conclusion of the study as it appears superficial.

Response 4 : We thank the reviewer for this comment. We have modified the paragraph to improve clarity.

Lines 439 “Obtustatin, Lebestatin and Viperistatin represent the shortest known snake venom monomeric disintegrins. In the present study, we used several in silico approaches, ranging from molecular modeling and docking to normal mode analysis, in order to understand the structural differences between three KTS-disintegrin molecules and their cognate receptor, the collagen IV-binding α1β1 integrin. Our study contributes to a better understanding of the structural distinctions between the different KTS-Disintegrins and their interactions with α1β1 receptors, which could highly improve the The first snake venom KTS/Disintegrins-integrin interactions using bioinformatics approaches”.

Minor Concerns:

Point 1: Was the DIα1 domain of Integrin fixed in docking? Have docking experiments been conducted at a specific binding site or have blind docking experiments been performed?

Response1 : Thanks to the reviewer for this thoughtful comment and effort towards improving our manuscript. The DIα1 domain of Integrin is fixed in this molecular docking. We would like to point out that before creating our docking, we conducted a bibliographical search for each probable interaction. This description is directly related to our previous work Khamessi et al., 2018 (RK, the first scorpion peptide with dual disintegrin activity on α1β1 and αvβ3 integrins) and Morjen et al., 2018 (Targeting α1 inserted domain (I) of α1β1 integrin by Lebetin 2 from M. lebetina transmediterranea venom decreased tumorigenesis and angiogenesis). In our study we propose a blind docking to cover the binding site of the collagen.

We have added this paragraph to improve clarity :

Lines 373 “Docking is performed over the entire surface of the integrin. The interface of the receptor used for this molecular docking study covers the binding site of the collagen”.

Lines 439 “Cluspro places the DIα1 domain of Integrin at the origin of the coordinate system on a fixed grid, the KTS-disintegrin is placed on a movable grid, and the interaction energy is written in the form of a correlation function”.

Point 2: There is no explanation of the full form of the terms KTS, RGD, KGD, MLD, WGD, and RMN.

Response 2 :

We thank the reviewer for this comment. The labels are now added in the current manuscript.

Lines 104 Arginine-Glycine-Aspartic (RGD), Lysine-Glycine-Aspartic (KGD) and R/KTS motifs, with respectively 40, 5 and 5 disintegrins (50 Disintegrins in total). The RGD group includes Methionine-Leucine-Aspartic (MLD) and Tryptophan-Glycine-Aspartic (WGD) motifs.

Point 3: Was the author interested in investigating compound features or mechanisms behind the interaction (line 14)?

Response 3 :

The ultimate goal of our study is to clarify the Lysine-Threonine-Serine (KTS) Disintegrins Integrin interaction model

Point 4: The title of Figure 3 indicates that the KTS motif is stained blue. I disagree with the use of the term "stained" here. A stain is a chemical dye used to color a specific chemical constituent for microscopy. As an alternative, I would call it a "KTS motif represented in blue."

Response 4 : We thank the reviewer for pointing out this error. It is now corrected in the current version.

"KTS motif represented in blue”

Point 5: A reference to the previous study is missing from the text (line 372).

Response 5 : We thank the reviewer for this comment. The reference is now corrected in the current version.

Reviewer 2 Report

The authors presented a good introduction which helps the reader understand this manuscript's proposal. However, the authors made some mistakes and were unclear in methodology, results, and discussion and I'll point out some issues hoping that they'll guide the authors to improve their manuscript quality.

- on pa.g 4 the authors presented on Fig. 2 the phylogenetic tree of Disintegrins and explaining that main motifs (RGD, KGD and R/KTS) represented three clades. A quick search on Uniprot database (or BLASTp) shows that are more integrin clades that could be included in this analysis. My suggestion is that authors could include other integrins for building a more robust phylogenetic integrin tree.

- on pag. 5 the authors show modeling results of KTS-disintegrins. In my opinion, Figure 3D could be improved using the multi-sequence alignment among all three disintegrins as described by ClustalW ou ESpirit3.0 sites. This could help the readers to compare these sequences identifying similarities and differences. 

 Figure 3A-C could be improved by changing the protein surface/ribbon/c-alpha sphere representation by cartoon (3D)/c-alpha sphere representation. It is abundantly cited on literature that many proteins with high sequence similarities presented highly diverse 3D structures. So, the authors should all 3D disitegrins structures generated by Modeller10.04

- on pag. 6 the Fig. 4 A and B presented some unclear results. The authors cited that C-terminals were flexible as noted in Fig. 4B, but the 23, 24 and 41 results were a little bit confusing since they are not shown in Fig. 4B. Another question: is it possible that one peptide (<50aa) presenting 4 disulfide-bridges present high or low mobility?

- on pag. 7, Tab. 1 presents Accessible surface areas of some amino acids. Is it possible that 21 to 24 aa sited on exposed protein surface (as Fig. 5A-C) had this little accessible area (<40%)? As C-terminal are highly exposed, all residues above 39 will have a high exposed area.

- on pag. 8, Fig. 5A-C show the prot-prot docking among Disintegrins and DIa1. All 3D results were kindly similar despite some minor aminoacids interactions. I judged that these figures will be more clear if the authors represent Disintegrins similarly DIa1, e.g. as cartoon.

Regards.

Author Response

The authors presented a good introduction which helps the reader understand this manuscript's proposal. However, the authors made some mistakes and were unclear in methodology, results, and discussion and I'll point out some issues hoping that they'll guide the authors to improve their manuscript quality.

Point 1: on page 4 the authors presented on Fig. 2 the phylogenetic tree of Disintegrins and explained that main motifs (RGD, KGD and R/KTS) represented three clades. A quick search on the Uniprot database (or BLASTp) shows that there are more integrin clades that could be included in this analysis. My suggestion is that authors could include other integrins for building a more robust phylogenetic integrin tree.

Response 1: We thank the reviewer for this comment. Vasconcelos et al., 2021 (Structure-Function Relationship of the Disintegrin Family: Sequence Signature and Integrin Interaction) suggested this current classification of disintegrins: long (84 amino acids and 7 disulfide bonds), medium (70 amino acids and 6 disulfide bonds), dimeric (67 amino acids and 4 intrachain disulfide bonds for each subunit and 2 interchain), and short (41–51 amino acids and 4 disulfide bonds). The disintegrins phylogenetic tree displayed in our study includes just the group of short disintegrins containing three clades corresponding to Arginine-Glycine-Aspartic (RGD), Lysine-Glycine-Aspartic (KGD) and R/KTS motifs.

We agree with the reviewer. We have modified the paragraph to improve clarity.

 “The disintegrins phylogenetic tree displayed in our study includes just the group of short disintegrins containing three clades corresponding to RGD, KGD and R/KTS motifs.”

Lines 102 “The disintegrins phylogenetic tree displayed in Figure 2, including the group of short disintegrins corresponding to Arginine-Glycine-Aspartic (RGD) group, the RGD like group [include Lysine-Glycine-Aspartic (KGD), Methionine-Leucine-Aspartic (MLD), Methionine-Glycine-Aspartic (MGD), Valine-Glycine-Aspartic (VGD) and Tryptophan-Glycine-Aspartic (WGD)] and the Arginine/Lysine-Threonine-Serine (R/KTS) group.

This short disintegrins family containing 41–51 residues and four disulfide bonds, has revealed three clades corresponding to (RGD), (KGD) and R/KTS motifs, with respectively 40, 5 and 5 disintegrins (50 Disintegrins in total).”

Lines 113 “Short disintegrins divided into groups, the color blue highlights the RGD group (24 disintegrins) and the RGD like group (14 disintegrins) includes seven MLD, five KGD, one MGD, four VGD and two WGD, Eristostatin group (green), KGD group (pink) and /RKTS (red).”

Point 2: on pag. 5 the authors show modeling results of KTS-disintegrins. In my opinion, Figure 3D could be improved using the multi-sequence alignment among all three disintegrins as described by ClustalW ou ESpirit3.0 sites. This could help the readers to compare these sequences identifying similarities and differences.

Response 2 : We agree with the reviewer. We have modified the figure.3D  to improve clarity (this is now corrected in the actual version).

Point 3: Figure 3A-C could be improved by changing the protein surface/ribbon/c-alpha sphere representation by cartoon (3D)/c-alpha sphere representation. It is abundantly cited on literature that many proteins with high sequence similarities presented highly diverse 3D structures. So, the authors should all 3D disitegrins structures generated by Modeller10.04

Response 3: We thank the reviewer for this comment. We have modified the figure.3A-C  to improve clarity and all structures have been generated by Modeller10.04 (this is now corrected in the actual version).

Point 4: on pag. 6 the Fig. 4 A and B presented some unclear results. The authors cited that C-terminals were flexible as noted in Fig. 4B, but the 23, 24 and 41 results were a little bit confusing since they are not shown in Fig. 4B. Another question: is it possible that one peptide (<50aa) presenting 4 disulfide-bridges present high or low mobility?

Response 4: We agree with the reviewer. We have added some paragraphs to improve clarity.

“Applications of NMA (normal mode analysis) cover wide areas of structural bioinformatics. ElNémo determines the normal modes that contribute the most to the corresponding protein movement when two conformations of the same (or a homologous) protein are available.

Lines 309 “Beyond Ser23 residue the profile shows us that this region is flexible and NMSD/R2 shows that some residues tend to be more mobile than others. Especially the residues in position Ser23, Arg24 and Gly41 this investigation has confirmed the direct involvement of its residues in the interaction with the target receptor. For example, residue 41 of Obtustatin exhibits high flexibility and the latter is involved in the interaction. Ser23 residue of Viperistatin and Lebestatin show high flexibility compared to Obtustatin. This residue is not involved in the interaction of the Obtustatin-DIl1 complex. Obtustatin lacks the Arg24 involved in the interaction and its predictions are perfectly correlated with the Fig. 4B. Thr22 residue doesn’t seem to intervene in the interaction and disintegrins molecular docking results are in accordance with the NMA. We would like to note that, it is possible to find flexible regions that contain stable residues or the reverse is due to protein conformations. This is one of the folding problems, for example the synthetic RGD motif (cilengitide) is composed of two residues R and D that are 100% exposed to the solvent. On the other hand, the residue G is not exposed to the solvent and even carries a percentage less than 20%, this is the case for several proteins especially the ones that carry an active site.”

For the last question: In fact, it is the specificity of the short family of disintegrins, even in the presence of 4 disulfide bridges, the peptides of this family (of 41AA) present AA which tend to impose a certain flexibility and this is confirmed by the fact that their 3D structure does not present either helix or beta sheet. The nature of the AAs between the cysteine does not favor the formation of secondary structures, which explains the presence of flexibility.

Point 5: on pag. 7, Tab. 1 presents Accessible surface areas of some amino acids. Is it possible that 21 to 24 aa sited on exposed protein surface (as Fig. 5A-C) had this little accessible area (<40%)? As C-terminal are highly exposed, all residues above 39 will have a high exposed area.

Response 5 : We appreciate the reviewer’s comment. We would like to note that, it is possible to find flexible regions that contain stable residues or the reverse is due to protein conformations. This is one of the folding problems, for example the synthetic RGD motif (cilengitide) is composed of two residues R and D that are 100% exposed to the solvent. On the other hand, the residue G is not exposed to the solvent and even carries a percentage less than 20%, this is the case for several proteins especially the ones that carry an active site. The positions of the cysteines that form the disulfide bridges directly influence the flexibility of the name residues within the same molecules. In our case the residues above 39 present a strongly exposed zone (in the whole region) because they are included in zones that are inside the disulphide bridges.

Point 6: on pag. 8, Fig. 5A-C show the prot-prot docking among Disintegrins and DIa1. All 3D results were kindly similar despite some minor amino acids interactions. I judged that these figures will be more clear if the authors represent Disintegrins similarly to DIa1, e.g. as cartoons.

Response 6: We agree with the reviewer. We have modified figure 5 to improve clarity.

Lines 192 Figure 5. Protein-Protein docking of KTS-family with domain I of the integrin α1β1. Three dimensional structures of complexes KTS-disintegrin showing the binding sites disintegrin with the DIα1 (green) and the main amino acid residues implicated in the interaction (¨Pink, Olive or Magenta). (A) Interactions formed by 6-amino acids of Lebestatin (Pink) within the domain I of α1β1. (B) Docking complex of Obtustatin-DIα1 showing that 5-amino acids of the Obtustatin (olive) interact with the residues (Green) surrounding the MIDAS site of the I-domain. (C). Interactions formed by 8-amino acids of Viperistatin (Magenta) within domain I of α1β1.

Reviewer 3 Report

the manuscript is well written ,some minor revision is required

1- Figure 2 requires better clarification

2- Figure 5 is not clear

Author Response

(The authors gave the same response as above.)

Round 2

Reviewer 1 Report

Thanks for making the corrections.

Reviewer 2 Report

Dear authors,  I have some concerns about the manuscript "The first snake venom KTS/Disintegrins-integrin interactions 2 using bioinformatics approaches".

 You presented a good introduction which helps the reader understand this manuscript's proposal. However, the authors made some mistakes and were unclear in methodology, results, and discussion and I'll point out some issues hoping that they'll guide the authors to improve their manuscript quality before the complete acceptance.

- on pag. 4 the authors presented on Fig. 2 the phylogenetic tree of Disintegrins and explaining that main motifs (RGD, KGD and R/KTS) represented three clades. A quick search on Uniprot database (or BLASTp) shows that are more integrin clades that could be included in this analysis. My suggestion is that authors could include other integrins for building a more robust phylogenetic integrin tree.

- on pag. 5 the authors show modeling results of KTS-disintegrins. In my opinion, Figure 3D could be improved using the multi-sequence alignment among all three disintegrins as described by ClustalW ou ESpirit3.0 sites. This could help the readers to compare these sequences identifying similarities and differences. 

 Figure 3A-C could be improved by changing the protein surface/ribbon/c-alpha sphere representation by cartoon (3D)/c-alpha sphere representation. It is abundantly cited on literature that many proteins with high sequence similarities presented highly diverse 3D structures. So, the authors should all 3D disitegrins structures generated by Modeller10.04

- on pag. 6 the Fig. 4 A and B presented some unclear results. The authors cited that C-terminals were flexible as noted in Fig. 4B, but the 23, 24 and 41 results were a little bit confusing since they are not shown in Fig. 4B. Another question: is it possible that one peptide (<50aa) presenting 4 disulfide-bridges present high or low mobility?

- on pag. 7, Tab. 1 presents Accessible surface areas of some amino acids. Is it possible that 21 to 24 aa sited on exposed protein surface (as Fig. 5A-C) had this little accessible area (<40%)? As C-terminal are highly exposed, all residues above 39 will have a high exposed area.

- on pag. 8, Fig. 5A-C show the prot-prot docking among Disintegrins and DIa1. All 3D results were kindly similar despite some minor amino acids interactions. I judged that these figures will be more clear if the authors represent Disintegrins similarly DIa1, e.g. as cartoon.

 Based on the descriptions above, I judge that this manuscript could be corrected before its complete publication.

Regards.